# Planning Perspectives and Approaches for Activating Underground Built Heritage

Carlos Smaniotto Costa [1,*], Marluci Menezes [2], Petja Ivanova-Radovanova [3,4], Tatiana Ruchinskaya [5], Konstantinos Lalenis [6] and Monica Bocci [7,8]

1   CeiED Interdisciplinary Research Centre for Education and Development, Universidade Lusófona, 1749-024 Lisbon, Portugal
2   LNEC-National Laboratory of Civil Engineering, 1700-066 Lisbon, Portugal; marluci@lnec.pt
3   CAWRI-BAS-Climate, Atmosphere and Water Research Institute at the Bulgarian Academy of Sciences & Association for Integrated Development and Sustainability, 1784 Sofia, Bulgaria; p.radovanova@gmail.com
4   Association for Integrated Development and Sustainability, 1408 Sofia, Bulgaria
5   TVR Design Consultancy, Willingham CB24 5JA, Cambridgeshire, UK; tvr281@hotmail.co.uk
6   Department of Planning and Regional Development, University of Thessaly, 38334 Volos, Greece; klalenis@gmail.com
7   Department of Agriculture, Food Sciences and Environment, Università Politecnica delle Marche, 60131 Ancona, Italy; monica.bocci@staff.univpm.it
8   Unione Montana del Catria e Nerone, 61043 Cagli, Italy
*   Correspondence: smaniotto.costa@ulusofona.pt

**Abstract:** This paper delivers actionable recommendations towards building a rationale for activating and promoting Underground Built Heritage (UBH) based on the nexus heritage, territory and society, and making use of existing literature and findings from five international cases. The research was conducted in the framework of the working group on Planning Approaches of the COST Action Underground4value. The analysis of the cases aims to provide guidelines for this working group and to benchmark good practices in activating UBH. It highlights the importance of community-led initiatives, leadership and dialogue and power sharing between the local/regional authorities and communities aiming for better understanding of the potential of UBH. The successes and/or failures of the five cases emphasise the importance of knowledge and experience in participatory approaches. Success was verified, when effectiveness and democratic principles were combined in the planning process, and local history is integrated with citizen science, co-creation and placemaking. The analysed approaches stimulate a new hybrid layer for activating UBH, provide mechanisms of mediation between people and heritage, and contribute to cultural and social dimensions of sustainability. This is a highly challenging endeavour, as it seeks to support and advance a sound understanding of UBH as a sustainable resource, backed by strategic stakeholder dialogue and contextual knowledge. Such effort requires a dynamic understanding of UBH values, knowledge, abilities and skills, towards creating more effective coalitions of "actors" within localities, by developing structures, which encourage long term collaborative relationships.

**Keywords:** underground heritage; placemaking; place and citizen-based approaches; public participation





## 1. Introduction

Underground Built Heritage (UBH), as defined by the COST Action Underground4value [1], is a unique cultural resource below the surface of the Earth, and includes natural and anthropic caves, underground burial/rites structures, mines and quarries, dwellings and infrastructures (cisterns, ancient drainage systems, tunnels, etc.) [2]. With the loss of its original purpose, an UBH asset often falls into oblivion and becomes part of a hidden and forgotten cultural landscape. UBH assets are often unexplored, not documented and underexploited [1]. The project Underground4value aims at systematising academic knowledge

and experiences on conservation, valorisation, management and promotion of UBH assets towards their sustainable re-use. One priority set by the project is to provide insights on the potential value of UBH to support the development of local communities. This entails the involvement of the concerned communities towards community-led approaches. In this context, the Working Group on Planning Approaches [3] aims at advancing knowledge in the interface of managing the transition and strategic stakeholder dialogue. In this interface, an approach has to take into consideration functional, social and cultural factors and legislative and regulatory frameworks. The process of activating and promoting an UBH is the transition of an undervalued asset to one, which provides a positive effect by harnessing community power and working with the community. Activating and promoting UBH is a highly challenging endeavour. As a transitional process, it requires a conceptual rationale which fosters innovative and integrative capacities. Thus, the purpose of this work is to provide the rationale for taking such a challenge by addressing a multidimensional framework to screen out sustainable approaches to value UBH. This rationale is based on two major premises: (a) The active engagement of all stakeholders involved and/or interested in UBH (central government and local authorities, scientific and technical personnel, economic agents and the public as organizations, groups or individuals), and (b) the impact of the planning and participatory processes on the success of UBH projects. The basic reasons for each premise are discussed, along with possible implications for planning approaches and policies.

The analysis of five cases related to UBH, highlights the most suitable participatory approaches, which combine effectiveness and democratic principles in the planning process. By examining particularities and challenges of each case and their impacts on the success or failure of the projects, the analysis suggests that the local history linked to the heritage and the social context is an effective way to increase intensity and scope of community engagement. It also confirms the importance of integrating place-based and citizen-based planning approaches—whereas the concepts of citizen science, co-creation and placemaking could provide guidance.

## 2. Framework

There is a great deal of discussion across Europe about achieving sustainability through valuing cultural heritage [4]. Thus, cultural clues must be linked to planning primers which in turn have to be extended into appropriate (planning) actions. In such an endeavour special heed has to be given to particularities of each underground asset, including their intangible qualities. Regardless of size or magnitude, a UBH asset is often something yet to be discovered and not profoundly obvious, but with potential of a "hidden treasure". Embodying such "something" grants a spatial dimension to civic participation and learning dimensions, which is explored in this paper.

The scope of civic participation in spatial planning is too frequently undertaken as simply an additional technique in the development process. Little thought has been given to the complexities of democracy, its theory and its practice, or to the sensitive issue of representation and the public interest in participation procedures [5] p. 3. Thornley [6] elaborated three alternative perspectives in defining participation in planning: (a) Consensus and Stability—the main goal of planning and participation is the elaboration of a plan which is effective, operational and technically sound, with limited negotiations but without conflicts between involved/affected parties. (b) Containment and Bargaining—the main goal of planning and participation is the elaboration of both, a plan which was collectively chosen out of a number of alternative scenarios, and a process which implements the plan and embodies social interactions and agreements between social partners. (c) Conflict and Increased Consciousness—emphasizing the creation of collective consciousness in people as it concerns their dominant and decisive role in planning their own environment. The production of the plan is not so much included in the main goals of planning, which is often used as means. Main actors in the planning process are local organizations, associations and in general organised groups of local citizens, which hold decision making

powers. In this framework, the intensity and scope of participation can be expressed in a two-dimensional system (Figure 1) in which, in the "x" axis shows scope, and number and variety of participation strategies initiated by the various categories of participants, measured from narrow (fewer) to broad (greater), and the "y" axis shows the intensity of participation [7] p. 19.

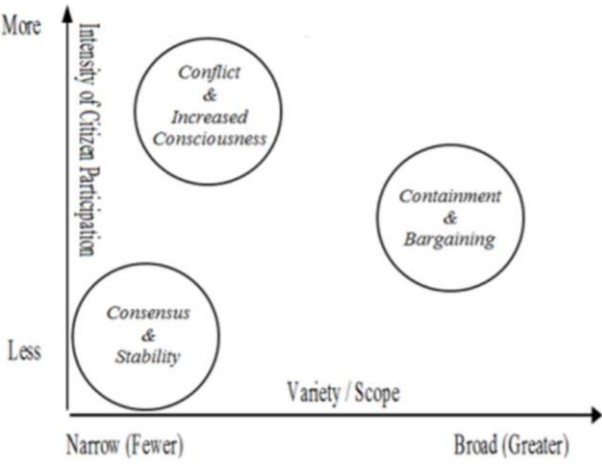

**Figure 1.** A proposed paradigm of citizen participation. Source: [8] p. 15, adapted from Cole [7] and author's elaboration.

In trying to position the three perspectives of planning and participation in this system, it can be concluded that: (a) "consensus and stability" is located in the area of low intensity of citizen participation and of fewer participation strategies, where conflicts between affected parties should be avoided. This is a technocratic approach in applied means, models and methods, with planners/scientists as the most significant agents. (b) "Containment and bargaining" are high on the scope and variety axis, while intensity might vary around medium. This perspective recognizes the existence of conflicts, but it attempts to integrate them through a series of adjustments and concessions. Authorities of various levels usually set the rules and supervise the bargaining process. The main goal is the implementation of the collectively developed plans and the achievement of agreements between partners. (c) "Conflict and increased consciousness" are high on the intensity axis, while scope and variety may vary. Here, conflict is seen as fundamental to the operation of society. The production of the plan is not so much included in the main goals, but the development of increased consciousness of citizens regarding their decisive role in planning their own environment [8].

In today's Europe, the Containment and Bargaining perspective is widely adopted in spatial planning since it gives importance to the engagement of a diversity of social actors from different positions. The role of central government and/or local authorities as arbitrators in this process is also recognized, setting the legal and regulatory framework and supervising the bargaining process. Finally, this perspective is the most suitable to combine effectiveness with democratic principles in the planning process, since it focuses on both, the elaboration and implementation of a widely accepted plan and the establishment of an ongoing process of social interactions [8] p. 13. The comparative advantage of bargaining is that it usually succeeds in ensuring public support and acquired expertise by many categories of participants. Citizen science, co-creation and placemaking are prime examples of Containment and Bargaining policies and may have a vital role in developing and managing UBH.

Five case studies from Bulgaria, Greece, Italy, Portugal and the UK provide valuable findings for this study. These projects showcase various types of participatory planning and approaches, which reinforced strategies for UBH.

## 3. Materials and Method

### 3.1. Cases Selection

In order to obtain a better understanding of the planning issues related to activating UBH, the WG participants were requested to bring their experience and knowledge to the table. This involved the description of UBH cases as well as the analysis of working methodologies and approaches, which could enrich the experiences. The cases have been chosen for (1) carrying a legacy of UBH and being related to local history, landscape and community; (2) presenting diversity, in terms of geographical distribution and experiences and (3) using a variety of methodologies and approaches to create a value.

As asserted by some authors [9–11] for a conceptual and methodological exploration, the variety of cases opens the opportunity to relate and compare individual cases with diverse conditions, traditions and circumstances and exploratory relationships [10]. The comparative strategy is therefore expected to provide insight on the range of partnerships and co-creation among academics and local communities towards generating new perspectives and functions to UBH assets.

### 3.2. Method

The WG Planning Approaches adopt a comparative case study methodology with an evaluation process to capture insights and to lay the foundations for a theoretical framework. The cases differ in terms of their maturity of reuse, communication strategy and involvement of stakeholders. The analysis of the five cases seeks to achieve transferability of knowledge and experiences while accounting for cultural and regulatory differences [9]. Built on these premises, the methodological approach is based on comparative urbanism [12,13], and thus focusing systematically on key processes, their different articulation and their inter-contextual interpretation rather than bringing cases into generalist umbrella or pre-assigned theories [11].

The five cases are discussed on the basis of a common framework, each case highlighting its own objectives, challenges met and the achieved results. In a second step the authors analysed the achievements and amalgamated these into particular topics for framing the discussion. These topics give the rise to issues, which are further discussed in Section 6 Building the rationale of activating UBH. This rationale is not conclusive as the Project Underground4value will collect and analyse further experiences.

## 4. Five Cases—Challenges and Achievements

### 4.1. Quality Assessment of Rila Monastery Nature Park, Bulgaria

This case investigates the assessment of the values of natural and cultural landscape and heritage of the Nature Park of Rila Monastery in Bulgaria with the Cave of Sv. Ivan Rislki Chudotvorets (St. John of Rila the Miracle Worker) which had been undertaken by the method of Rapid Landscape Assessment (RLA) [14].

The cave St. John of Rila is located in the Nature Park (25,000 ha) owned by the Bulgarian Orthodox Church and listed as a UNESCO World Heritage Site. The cave, which can be reached by a steep path starting from the Rila Monastery, is a sacred place where St. John of Rila lived during his monasticism. St John is the most important Bulgarian hermit saint of the Bulgarian Orthodox Church, who was revered as a saint while he was still alive. There are many legends surrounding him and many churches are founded in his honour, including the Rila Monastery.

The RLA, as part of the Rapid Ecological Assessment (REA) of the Rila Monastery Nature Park, was funded by The United States Agency for International Development (USAID) within the Biodiversity Conservation and Economic Growth programme and conducted by an interdisciplinary team of experts in natural and social sciences, representatives of central government, local institutions, Bulgarian Orthodox Church, small business and NGOs [15].

The RLA framework was based on basic concepts and principles of landscape planning and conservation [16], and assessment of the quality and visual appearance of the

landscapes, which affects human psychology in a variety of ways [17–19]. The factors and indicators of quality assessment of the cultural, underground heritage and landscape had been discussed in participatory workshops and the final set of indicators were ranked according to their importance. Several field trips and follow-up laboratory work had been organized by a multidisciplinary team, including experts in urban planning, landscape design, forestry, sociology, conservation, local governance and tourism. The assessment included evaluation of several categories: natural resources (i.e., rock formations, forests, meadows, aquatic elements, plant and animal species), cultural, historical and underground landmarks and infrastructure sites; the quality of landscapes (i.e., picturesque, natural, landscape diversity, vulnerability, and accessibility); and visitors' psychological response. RLA also included a socio-economic survey [20], which assessed the non-monetary benefits of the park (e.g., aesthetic enjoyment, comfort, and safety).

The RLA of Rila Monastery is a clear case, where the assessment provided the main input to the management plan which is open to the general public. This is also an example of good practice in participatory framework and successful development of a management plan with recommendations for zoning of the area for both environmental and economic improvements of the park according to the quality of the natural and cultural landscapes.

*4.2. Archaeological Excavations in Thessaloniki Metro, Greece*

A metro system for Thessaloniki has been a controversial issue since the beginning of the 20th century, but the work started only in 2006. During the excavations, a large number of important archaeological findings was discovered. In particular, in Venizelou station, the 75 m long and 5.5 m wide, Roman Decumanus Maximus road, also called Byzantine Middle Road of Thessaloniki was revealed, built by the Roman emperor Galerius in the 4th Century A.D. and reconstructed two centuries later. Next to the Middle Road, a four pillars gate was also discovered, highlighting the most important crossroad of the city at the time, with the pathway named Cardo (Figure 2). This spot essentially marked the commercial heart of the Roman and later, the Byzantine city. Notably, the same crossroad is still considered as one of the most important points of the modern business centre [21]. This monumental ensemble, with public areas and buildings, wide streets, brilliant in its original construction, paved with marble and with rectangular stone slabs, demonstrates over a large area the public space structure of Thessaloniki—the second most important city of the Byzantine Empire—also known from other Byzantine cities but never revealed in such a large scale and a good state of preservation. So far there is no comparable complex in other Byzantine cities, not even in Constantinople, that features in such clarity the monumental urban planning and structure dating from late antiquity up to the transitional period and the beginning of the Middle Byzantine times (4th–9th century AD). Furthermore, the Venizelos Station archaeological complex is historically and culturally related with an ensemble of 15 monuments of Thessaloniki, which have been designated, in 1988, as World Heritage sites and were included in the UNESCO List [22]. A treasure trove of 750 jewels and more than 2700 burial artifacts were also discovered in other metro stations currently under construction, all testifying to the heritage of Hellenistic and Roman times. International archaeological circles have since characterized the discovery as "Byzantine Pompeii" [23].

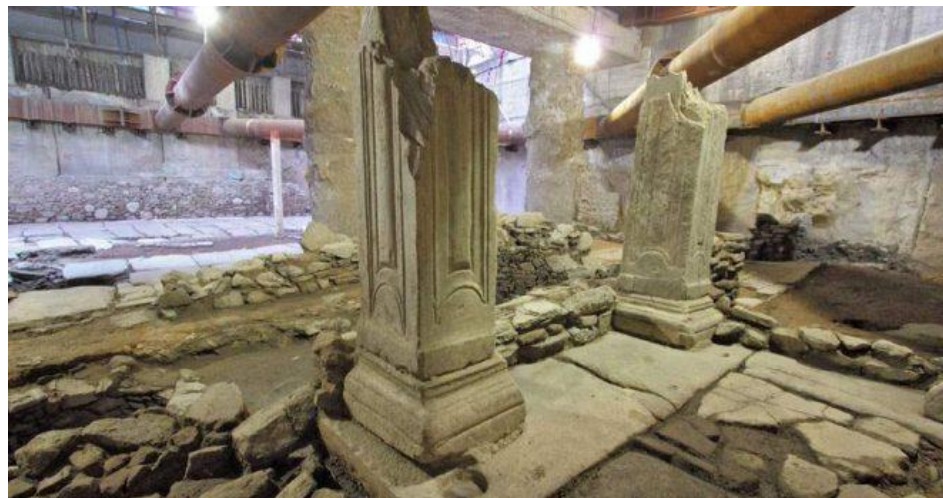

**Figure 2.** Thessaloniki, Venizelou Street Station: street intersection and tetrapylon. © Ministry of Culture and Sport: 9th EBA (free use of photos).

Because of this discovery the metro line was redesigned, with tunnels at a depth up to 31 m—as opposed to the initial 8 m—and providing for mini museums in the stations [23,24]. By February 2019, the main line construction was completed by 95% [25,26]. In September 2019, Greece's new conservative government announced the decision to scrap the previous plan to keep the archaeological discoveries in situ, and move the metro lines deeper, choosing instead to disassemble and reassemble the antiquities at a later stage, in order to reduce excavations costs. As main reasons, from the technical point of view, it was claimed that the construction process would put in situ antiquities in danger and the safety in the metro operation on the deepest level was not guaranteed. Criticism against this decision focused on the undemocratic way that the government imposed its decision on local communities, on the practical impossibility to reassemble roads, walls, gates in such a scale and with unclear method, and on the lack of studies on how artefacts could be returned and re-assembled once the station has been built. Furthermore, the two years' delay of completing the metro, and the new, out of contract, works of disassembly, will grant the construction company compensation of several million Euros [26,27]. Controversy was also caused by the replacement of several members of the Central Archaeological Council (KAS) by the Ministry of Culture, in order to extract a majority vote in KAS for the transfer of the antiquities—contrary to two previous decisions for keeping them in-situ [21,24], by stating that "the Venizelos Station antiquities are considered as a whole, forming an important archaeological site and a unique example of the Byzantine urban space preserved in situ and a solid testimony, at international level, to the function of Byzantine cities" (session No2/15-01-2013). Cultural groups, local organizations, the Aristotelian University of Thessaloniki, and archaeologists worldwide protested against the government's decision on legal, technical and social grounds [23,28]. Byzantinology scholar Paolo Odorico, director of the Center for Byzantine and Modern Greek Studies at Ecole des Hautes Etudes en Sciences Sociales in Paris, visited the antiquities of Venizelos Station in 2013. He argued that the displacement of the Byzantine Middle Road would irreversibly destroy the structural authenticity of the antiquities of the monumental set, a value protected by international and Greek law [23]. Demonstrations were organized, information meetings and scientific conferences analysed the technical disadvantages of the governmental decision, and petitions were signed [25,26]. Appeals were submitted in 2020 by the Hellenic Society of Environment and Culture together with the Christian Archaeological Society, by 26 Thessaloniki citizens, and by the Association of Greek Archaeologists together with another 10 institutions. In the appeal signed by its Executive President Prof. Dr. Hermann Parzinger, Europa Nostra highlights the "European significance of the aforementioned archaeological remains, which constitute a unique find of an urban

ensemble of monumental scale and in a very good state of preservation. It is the only one preserved in the whole area once occupied by the Eastern Roman Empire and dating to the relatively unknown Late Roman and Early Byzantine period (4th–9th century AD)" [26]. The appeals were rejected by the court, on 29 June 2021, with a paper-thin margin of 13 out of 25 votes among the panel members. In August 2021 works for the displacement of the antiquities started in a slow pace, while reactions continue, by internationalization of the issue, appeals of international organizations to the Greek Prime Minister to keep antiquities in situ, organization of tours to the sites of antiquities, presentations in the Municipal Council, and collection of signatures calling for immediate stop of the displacement of antiquities [25].

### 4.3. Building Collective Memory on Cabernardi Sulphur Mine, Sassoferrato, Italy

During the first half of the 20th Century, Cabernardi mine was the largest sulphur mine site in Europe with a production of 60,000 tons of sulphur [29]. The mine was approximately 8 km long, 1500 m wide and 800 m deep, with 21 tunnels and two extraction wells at 460 m. At its peak of operation, it employed 1600–1800 miners who lived in a settlement called Cantarino Village built by the Montecatini Society, the national owning company. Cantarino Village was completed in 1919 and provided necessary infrastructure and facilities including a school, playgrounds and recreational areas [30]. The mine was closed in 1959.

In 1983, a group of ex-miners and/or their descendants decided to revive the tradition to celebrate in-situ Santa Barbara's day, the patron saint of miners. They wanted to remember those people who have died working in the mining industry during the years of its operation. This opened the opportunity to collect photographs and objects related to working in the mines, which were displayed in an exhibition. In 1992 a museum was opened, with an exposition of minerals and objects donated by miners and their families. Since 1997 the cultural association "La Miniera Onlus" is managing the museum and a mining/archaeological park (Figure 3). People still donate objects of miners to the museum (i.e., guardian hats, workbooks, boots, and lamps). In April 1999, the museum was moved to its present location, in the old primary school building, in the wake of a collaboration between "La Miniera Onlus" and the Sassoferrato Municipality.

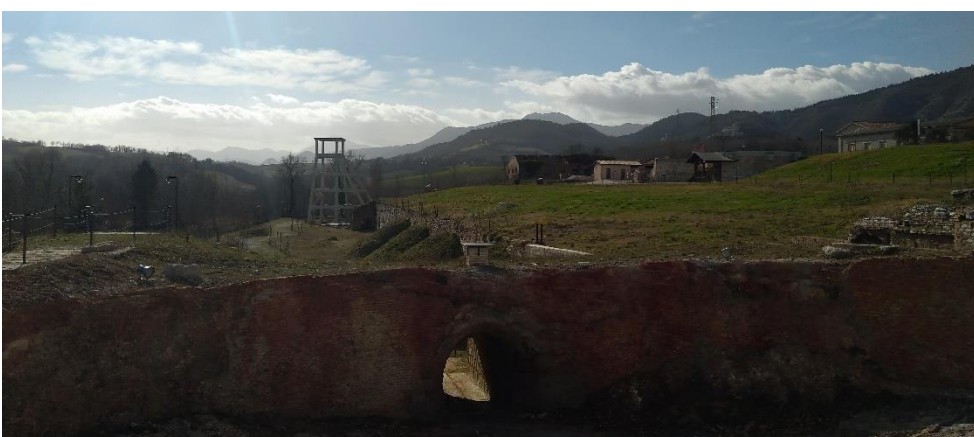

**Figure 3.** A panoramic view of the Cabernardi Mine Park. Photo: Bocci, 2020.

Thanks to a recent restoration, most of the above-ground structures in the mining park can be visited, including the steam-electric power plant, the service tunnel and the Donegani well. Further accessibility to other parts of the mine is on the agenda, but interventions to value the site require significant financial resources. Nevertheless, many initiatives are in place to promote the site as one of the landmarks in central Italy.

### 4.4. Public Art Gallery in Lisbon Subway Stations, Portugal

Since 1959 the Metro de Lisboa (ML—Lisbon Subway) has been a key element of the mobility system in Lisbon. By 2020 the network expanded gradually to a length of 44.5 km, with fifty-six stations and four lines, with connections to other means of transport. ML is a government-owned company promoting economic, environmental and social sustainability of the metro system and the surroundings of the stations. Social sustainability focuses on giving special heed to stakeholders and using strategies that enhance gender equality, inclusiveness, urban resilience, innovation and climate change mitigation and adaptation strategies. As such the ML Art Project aims to turn subway stations into underground public art galleries and get stakeholders involved in decisions on enhancing the customer experience. Efforts for encouraging ridership, improving travel comfort and users' well-being go together with "mitigating" the transition between the surface and the underground environment [31]. Decorative tiles (azulejos) are particularly relevant in the project, as they are a common element in the Portuguese architecture and are used in the project in order to enhance the level of familiarity with the place and overcome the scepticism of descending underground. Thus, some stations provided tile installations on the walls, and others have colourful staircases. ML offers guided tours (with touristic, technical and study backgrounds) to the subway collections, publishes thematic books on underground art and provides support for scientific studies. The ML Art Project underlines strategies that enhance participatory processes and get stakeholders involved in the generation of innovative ideas. The ML strategy, coupled with sustainability goals and social responsibility embedded in formal planning and involving individual and collective coping, open an excellent opportunity to reflect on potentials and challenges of art to change the way people experience the city (over and under the surface).

The complexity of urban environment, transport and mobility systems, the scale and multi levels of the metro stations and the design of public spaces surrounding the stations conflict with human scales; they do not help people feel well in the transition between the surface and underground strata. The introduction of human scale art objects and traditional art familiar to the public in the subway enhances spatial trust, pleasantness and appropriation of ML.

### 4.5. Stabilisation of Underground Mines at Combe Down, Bath, UK

Two centuries of mining Bath limestone left a huge underground void beneath the village of Combe Down, a major suburban area of Bath. The mines were abandoned in the mid-1800s. In 1999 it was detected that the roof and pillars of the old mines were collapsing. The project task was to mitigate the potential risk to life and property in a manner that also accommodated major environmental, heritage and social concerns [32]. The threatened area was estimated to be 25,608 ha, including over 600 homes and infrastructure. The Combe Down project initially had challenges. Its plan area was unknown, the mine's void height and means of stabilization were uncertain. There were also three major statutory environmental constraints:

- The mine was home for large numbers of rare and legally protected bats.
- It lies in the groundwater Inner Source Protection Zone of a public water supply.
- There was legally protected archaeological heritage within the mine itself.

The project was funded by the Land Stabilisation Programme, operated by the Homes and Communities Agency (formerly English Partnerships). The client, Bath and North East Somerset Council, showed a clear leadership, advised by appointed consultants and initiated an extensive communication plan. A Steering Group was set up to inform stakeholders on project progress and receive their feedback. The success of the project was largely due to the extensive partnerships and the involvement of the community. An established Community Association had influence on all project decisions. Council was sharing power with the community, training the public to plan, execute, analyse and make decisions, as means of incorporating the public into the project. At the same time, it appointed professionals to liaise with consultants. The stakeholders' interests were not

aligned with the values of community economic survival in the beginning of the project. For example, English Heritage and Natural England wanted to protect rare bats species, living in the open galleries. The residents and insurers were looking for a complete infilling to secure homes.

The knowledge about underground conditions, and the requirements of different stakeholders were considered and combined in order to find the extent of compromise needed, and this provided a framework to the project strategy. As a result, it was decided to stabilize only the collapsing galleries with use of low-density foamed concrete.

Regular risk workshop events, public art, a comprehensive website and a Project Information Centre were set up to give extensive information and to solicit views. During the workshops, a quantitative weighting system to each factor influencing the scheme was used followed by discussions on relative allocation of points. The difference between critical factors and high-profile factors with less importance made clearer picture.

Research and consultation at the time (2004) indicated a need for a "contingent valuation" approach to be used for ascribing a monetary value to archaeology. A proxy approach was used for quantifying the economic impact of losing/retaining the archaeology by estimating its non-use value, that people derive from the knowledge that the site exists, even if they never plan to visit it. This assisted to build a greater consensus amongst stakeholders. The history of the community was used as an effective way to increase community engagement and participation in the project.

## 5. Planning Perspectives, Approaches and Bottlenecks: Framing the Debate

The five analysed cases present different ways of activating UBH. Driven by different conditions, policies, government regulations and community expectations, they portray values of underground/cultural assets as well as political awareness about their importance and potentialities. As stated in the Section 3.2, in reference to the method of analysis, the five cases are examined according to specific dimensions constituting a common framework. Through this examination, each case highlights its own objectives, challenges met and the achieved results. These specific dimensions are: 1. the intensity and scope/variety of participation of agents involved in the UBH cases, 2. the role of authorities of various administrative and/or political level in setting the legal and regulatory framework for managing UBH and organizing/supervising participatory and bargaining processes among stakeholders, 3. the role of the public/citizens in the form of organizations, NGOs, associations, groups, individuals etc. in initiating, participating, bargaining, co-creating and managing UBH policies and projects, and 4. the means used by all the involved agents for their involvement in UBH projects.

Intensity and scope of public participation was explored by positioning the five cases in the two-axis system analysed above (Figure 4). The case of Thessaloniki shows how the local community adopted a "conflict approach". It increased public participation in order to oppose the undemocratic decision of the central government, which they considered as harmful to local heritage. It is notable that this "reactive" strategy is complemented with a "proactive" one, trying to solidify its position for the in-situ handling of the antiquities, with technical and scientific virtues. The Thessaloniki case, thus, should be divided in two perspectives on planning and participation. The perspective of the central government (Thessaloniki 1) lies in the domain of consensus and stability with minimal participation processes, where the policy choices are solely based on economic and technical reasoning. The perspective of the public (Thessaloniki 2) on the other hand, exhibits a high degree of intensity of participation ("reactive" strategy), while it also scores in the Scope/Variety axis ("proactive"). Limitations to the latter were imposed by the opposing authorities, in terms of lack of finances and legitimacy. In this context, the "power/influence" question arises, in comparing/relating strategies and final outcomes to the objectives and the characteristics of each participant (local community vs government).

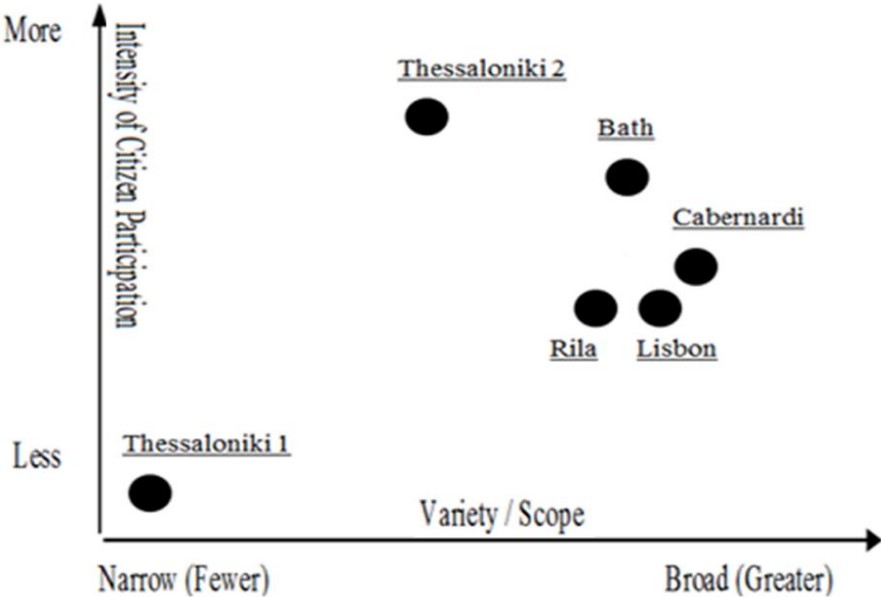

**Figure 4.** Perspectives of case studies on planning and participation. Source: [8] p. 15, adapted from [7] and authors' elaboration.

Bath, Lisbon, Cabernardi and Rila Monastery cases, on the other hand, showcase the "containment and bargaining" perspective where the cooperation between stakeholders and authorities was sought and/or adopted.

In the Bath case, the risk of a disaster triggered the initiation of the UBH project and encouraged the intensity of public participation, as the initial expression of conflicting interests of various stakeholders. At this stage, the communication plan and a co-creation technique proved to be successful in finding solutions acceptable for all participants. A distinction was made between co-created knowledge on UBH and co-created final value (means of stabilization). The knowledge about UBH and the technical knowledge were co-created first, in order to study risks and find the extent of compromise needed, in accordance with the requirements of the different stakeholders. This knowledge was able to provide a commonly accepted framework of the project strategy, and create "common value".

The role of central/regional/local authorities in setting the legal and regulatory framework and supervising the bargaining process was evident in Bath, Lisbon and Rila Monastery. In Bath, the regional authorities successfully played the role of arbitrators, setting the regulatory framework, and supervising the bargaining process. In Lisbon, the government owned company was the initiator in organizing and enhancing public participation by adding the dimension of an exploratory lab for citizen science. In Rila, the central government initiated a quality assessment process, where public participation was expressed in the form of cooperation of multidisciplinary participating agents. In this particular case, the assessment of psychological values of the place was combined with statutory assessments, ensuring that the public is given early and effective opportunities to participate and value the UBH.

In the Cabernardi case, the initiative for the UBH project came from the citizens and the local authorities who were eager to join in and provide support. Cabernardi mine was kept closed for a long time before the group of ex-miners and their descendants decided to celebrate in the mine the day of the patron saint of miners. By then, it became evident that the mine could offer many cultural and economic opportunities. The capacity of citizens to "make things happen", proved, in this case, to be more effective than the design of any scheme or support network [33].

Different initiatives such as setting up a museum (Cabernardi), enabling experiences for tourists (Rila Monastery) and Subway Art Project (Lisbon), bring elements of the every-

day life to UBH, opening an excellent opportunity to change the way people experience the "invisible" parts of the city, using the history of the community as an effective way to increase scope and intensity of community participation.

## 6. Building a Rationale for Activating UBH

Sustainability has been the main target of European policies concerning the natural and manmade environment during the last decades, and cultural heritage is considered one of the main dimensions of it. This acknowledgement is, however, carried mostly on a theoretical level, as Nocca [34] aptly points out. The author calls for producing evidence about the contribution of heritage to improve economic, social and environmental productivity. It is evident that local history linked to heritage and its social context is an effective way to enhance community interest and foster public participation in related plans and projects, such positive correlation has been also observed in other cases [34–36]. UBH with unique characteristics constitutes a significant part of heritage. Regardless of size or magnitude, UBH, as explained above, has the potential of a "hidden treasure", which is yet to be discovered and explored. Embodying such "treasure" grants some unique dimensions of UBH and new values because it is not a familiar part of citizens' everyday life and therefore it attracts public interest. A growing interest in these novel and hidden assets is acknowledged [36,37] for learning, recreational and tourism purposes [34,38].

This out-of-the-ordinary condition adds an important learning dimension to civic participation, scoring high, thus, to the scope and variety of the two-axis system analysed above, while maintaining the management aspiration of the public adds the important learning dimension. There are increased opportunities for cases of planning and management of UBH, located in the "Containment and Bargaining domain". The analysis of the five cases validated the theory that this perspective is the most suitable to combine effectiveness and democratic principles in the planning process, since it focuses on the development of a mutually accepted plan and its implementation, encouraging social interactions and agreements between all participants [8] p. 13. However, conflicts of interests of different stakeholders are likely to arise due to the nature of UBH. In the analysed cases, if conflicts are not resolved and intensity of participation is increasing, the prevailing perspective is "Conflict and increased Consciousness", where the "power/influence" factor arises, and in the planning outcome there is a "winning side" and a "losing side". If, on the other side, conflicts are recognized and resolved through bargaining and mutual concessions—as containment suggests—solutions acceptable to all parties would be adopted.

In this perspective, the initiatives in activating UBH should be expressed through clear understanding of the project requirements, making it transparent and creating an open environment to meet those requirements in the most effective way. Such demands are also made in regard to making places [38] where creativity is part of the process of weaving intangible factors to be found in atmospheres and activities together with the involvement of the community. The analysis of the five cases shows that the knowledge about UBH, requirements of different stakeholders and technical knowledge should be co-created first in order to understand potentials and risks and find the extent of compromise needed. A weighting system to each factor influencing the scheme is proved to build greater consensus amongst stakeholders and reach decisions.

The success of cases highlights the importance of dialogue and power sharing between the local/regional authorities and the local communities, where the sensitivity to local context is a basic foundation [33,34,37]. The driven dialog contributes to a collective behavioural change, and the community representatives get empowered to influence on project decisions. This also calls for the provision of organisational/governmental structures to strengthen local groups and provide practical/technical support. Such greater local empowerment has been part of the European Union Cohesion Policy, reinforced in its current multiannual framework [39]. A communication plan, defining how to get project information to people, should provide a framework to involve stakeholders in decision-making process and co-creation tools may assist in finding commonly acceptable

solutions. Several authors/researchers, besides calling attention to the matters of an effective communication plan, have also detailed the range of activities and strategies which such a plan should tackle, e.g., PPS [40], UrbAct [41].

The combination of place-based and citizens-based approaches ensures a deep understanding of the complexity of relationships between cultural heritage, social practices and technical and economic factors, and of the ways that they are/can be mobilized. This should establish innovative interactions, reflecting the result in a mutual and continuous value creation process (Figure 5).

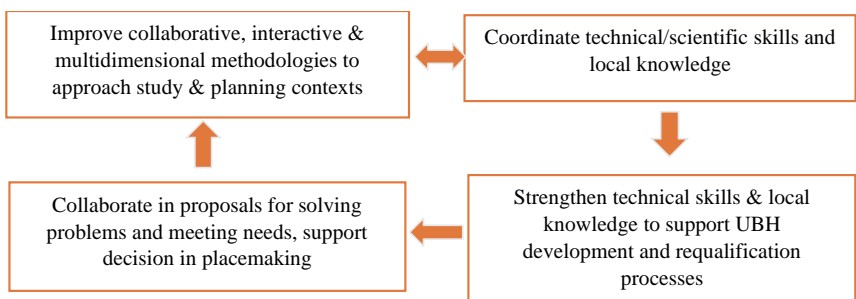

**Figure 5.** The dynamic contribution of citizen science and co-creation in a place-based approach. Source: [42].

This process evolves promoting local knowledge and citizen science in placemaking. Placemaking, as a bottom-up community building approach, is closely associated with citizen science, co-creation, and community engagement activities. In turn, placemaking is put in place through living labs, as it facilitates the creation of solutions allowing different actors to design, test and learn from socio-technical innovations [43]. Communities' appropriation and attachment of meaning and values to a space turn a space into a place where the physical, social and mental dimensions become relevant [44]. Placemaking, as advocated by the Project for Public Spaces [40] is a collaborative process towards reshaping the public realm to create shared values and bringing together different stakeholders in co-creating knowledge. Checking all the dimensions calls for establishing close interconnections between:

- Key stakeholders (including identification of their main characteristics, interests and potential forms of collaboration).
- Types/forms of representations of individuals/groups (considering the heterogeneity of practices and socio-demographic characteristics).
- UBH assets (including identification of the asset, its surroundings and its interrelations (i.e., between the private and public spaces) and considering spaces to be involved and mobilized in the intervention).

Local history, leadership, dialogue and power sharing are basic drivers for activating UBH, although these are not exclusive of UBH as Nocca [35] reports, they concern various kinds of heritage. UBH has a number of other cross-cutting issues, due to its nature. UBH projects have to face a lot of unknowns. For this reason, flexibility in planning approaches to enable future changes can be particularly useful. In this respect, authors that challenge flexibility call attention to the use of creativity [35,38,40] and in this process digitalisation could be an important means to ensure greater access to spaces and heritage [35,38,44]. Through this lens, as some researchers propose [34,35,37,38], creativity is an important way to boost cultural tourism. In the end, creating a tourism destination is an attractive way to preserve UBH and to provide benefits to the community [1,36].

Furthermore, in the decision-making process a greater weight should be given to long term environmental impacts [14]. In particular, impact assessments can be successful in looking into the multiple functions of UBH and its synergic action, while incorporating them into the early planning stages of projects to mitigate delays, logistical problems and reputational risks. To conduct such complex assessments, it is necessary to review

technical, planning and design proposals, together with past interventions, and define a technical-methodological procedure, which will safeguard cultural, environmental and underground heritage. The main challenge here is to identify logics, mechanisms and tools that support this approach.

## 7. Concluding Remarks

An open, procedural and contextualized approach to UBH, which incorporates the nexus between heritage, territory, society, culture and intervention, is an innovative strategy as it forms the foundation of initiatives to activate UBH. A planning rationale backed by the articulation of local and scientific knowledge encourages multidimensional, interactive and sustainable intervention dynamics.

The cases highlight the importance of leadership, dialogue and power sharing between the local/regional authorities and communities. Leadership is associated with several aspects including communication between stakeholders, managing conflicts of interests as well as the successful planning, which should comprise a combination of a place-based and citizens-based approaches. This is in accordance with the assertion of par. 2 above (Framework) that the Containment and Bargaining perspective is widely adopted in spatial planning in Europe, since it recognizes the role of central government and/or local authorities as arbitrators while it also gives importance to the engagement of a diversity of social actors from different positions. In the examined cases it also appeared that it combined effectiveness and democratic principles in the planning process, where local history linked to heritage and social context functioned as an effective way to increase intensity and scope of community participation.

Conflicts of interests of different stakeholders should be recognized in UBH projects and can be resolved by a framework, which involves all stakeholders in decision making processes (e.g., communication plan) and methodological tools (e.g., co-creation, citizen science and place-based approach). The combination of these approaches helps to build greater consensus amongst stakeholders and reach mutual decisions.

Prioritizing social, economic, environmental and cultural sustainability helps to deal with conflicting interests of the stakeholders. The dynamics/inertia of formal planning procedures and management structures can be increased with flexibility in the production and social use of the territory. In this context, the borderline between formal and informal planning stimulates a new hybrid layer for activating UBH, which provides a mechanism of mediation between people, heritage and development and helps to define the cultural and social dimensions of sustainability.

**Author Contributions:** Conceptualization, C.S.C., M.M., P.I.-R., T.R., K.L. and M.B.; methodology, C.S.C., M.M., P.I.-R., T.R., K.L. and M.B.; validation, C.S.C., M.M., P.I.-R., T.R., K.L. and M.B.; formal analysis, C.S.C., M.M., P.I.-R., T.R., K.L. and M.B.; investigation, C.S.C., M.M., P.I.-R., T.R., K.L. and M.B.; resources, C.S.C., M.M., P.I.-R., T.R., K.L. and M.B.; data curation, C.S.C., M.M., P.I.-R., T.R., K.L. and M.B.; writing—original draft preparation, C.S.C., M.M., P.I.-R., T.R., K.L. and M.B.; writing—review and editing, C.S.C., M.M., P.I.-R., T.R., K.L. and M.B.; visualization, C.S.C., M.M., P.I.-R., T.R., K.L. and M.B.; supervision, C.S.C.; project administration, C.S.C.; funding acquisition, C.S.C. All authors have read and agreed to the published version of the manuscript.

**Funding:** This paper has been supported by the funds of the COST Action CA18110 "Underground Built Heritage as Catalyser for Community Valorisation", https://www.cost.eu/actions/CA18110 and https://underground4value.eu (accessed on 26 August 2021).

**Institutional Review Board Statement:** Not applicable.

**Informed Consent Statement:** Not applicable.

**Data Availability Statement:** Data are contained within the article or available from referenced sources.

**Conflicts of Interest:** The authors declare no conflict of interest.

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
