# Peer review of "Planning Perspectives and Approaches for Activating Underground Built Heritage"

_sustainability, doi:10.3390/su131810349_

Round 1

Reviewer 1 Report

The paper concerns a very interesting study on Underground Built Heritage, presenting 5 case studies from relevant countries. In order for the scientific soundness of the paper to be enhanced, some aspects should be taken into account by the authors, as following:

The term ‘rationale’, repeated in sections 1 and 2, should be better replaced by other terms as ‘strategy’, ‘integration plan’, ‘principles’ etc.

In section 1, a description of UBH should be made, categorizing the key assets it may concern. Relevant References should be given. Respectively, section more data regarding UBH problematics, principles and integration strategies should be added. Good practices from literature should be also given. Additionally, the framework of this study should be described, including the collaboration means of the consortium, the selection of the case studies, as well as its scope and perspectives.

Section 2.2 concerns a chronicle of the actions, activities and protests implemented during the metro excavations in Thessaloniki. There is no reference on the great antiquities excavated and restored during the metro construction works, the integration plans assessed, while there are no relevant scientific references. This section should be reorganized on scientific criteria, emphasizing on the underground heritage discovered in the city of Thessaloniki due to the metro construction works.

In section 3, lines 270-272, it is mentioned that ‘increased public participation in order to tackle the lack of institutional support’. This part should be revised, taking into account scientific data.

In section 4, relevant references from literature should be given, in order to better correlate the research findings of the study.

Author Response

To Reviewer 1

Thank you for our remarks and suggestions!

  • “rationale” is kept, but we provided a more consistent explanation of the building process.
  • Description of UBH was added - we supposed this was redundant as this text is submitted to a special issue.
  • More data regarding UBH problematics, principles and integration strategies was added. Additional good practices from literature were not given, considering that the number of cases examined in the text was already high, and any further addition with more cases would be dysfunctional. This paper is also part of a special issue: ‘Going Underground. Making Heritage Sustainable’, other papers will provide for sure further examples.
  • Added is a description on the selection of cases and the methodology ( Materials and Method)
  • The Thessaloniki case was elaborated with references to the excavated antiquities. Further details on technical issues would divert the reader from focusing on the issue of democratic and participatory management of UBH, to issues of technical nature.
  • Lines 270-272 were revised and rewritten.
  • The new inserted text parts are marked in blue.

Reviewer 2 Report

Authors,

This is an interesting topic, and the selection of review of case studies makes for good reading. However, the data are quite skimpy and claims for 'proof' aren't based on much empirical data. I would urge the authors to dig in more strongly on what is unique to UBH and why focusing on public participation in addressing UBH resources merits scholarly attention.

I look forward to seeing a revised manuscript.

More detailed comments are attached.

Author Response

To Reviewer 2

Thank you for our remarks and suggestions!

  • UBH was defined and described. Examples were not given, since the number of UBH cases examined in the paper is already high. This paper is also part of a special issue: ‘Going Underground. Making Heritage Sustainable’, other papers will provide for sure further examples.
  • Fig 1 was corrected.
  • The case for UBH has been elaborated and its uniqueness has been exhibited.
  • A description on the selection of cases and the methodology ( Materials and Method) is added to the text.
  • The Thessaloniki case, which is more complex in comparison to the other ones, was expanded. The comparative approach of analysis of the 5 cases was stressed, and there the common evaluative criteria were described. Fig 2 is useful because it visually facilitates the related discussion which correlates the dimensions of participation in each case with the theories of social order, and the classification of the 5 cases in these theories.
  • The key words in quotes, in lines 270-272 are explicitly described in the Framework section. A reader who has read this section with a moderate degree of attention shouldn’t have any problem in comprehending the terms. Furthermore, these terms (“words in quotes”) have been established in international bibliography as such, so there are no “correct” words to describe them better.
  • The term “proved” has been replaced by a sentence showing what the outcomes of the analysis were indicating, in accordance with a hypothesis that is clearly stated before the sentence in question.
  • Fig 3 and jargon - the figure 3 is kept as it illustrates well the outcomes; re jargon - these are the outcomes of the analysis. However, further literature is provided to substantiate our arguments.
  • The claim of proof was eliminated and replaced by indications stemming out of the case studies.
  • The new inserted text parts are marked in blue.

Round 2

Reviewer 2 Report

Dear authors,

Thank you for addressing my concerns. I look forward to seeing this research published in Sustainability shortly.